# Does adding MRI and CSF-based biomarkers improve cognitive status classification based on cognitive performance questionnaires?

**Mateo P. Farina**[1,2]*, **Joseph Saenz**[3], **Eileen M. Crimmins**[1]

**1** School of Gerontology, University of Southern California, Los Angeles, California, United States of America, **2** Human Development and Family Sciences, University of Texas at Austin, Austin, Texas, United States of America, **3** Edson College of Nursing and Health Innovation, Arizona State University, Phoenix, Arizona, United States of America

* mateo.farina@austin.utexas.edu

## Abstract

### Background

Cognitive status classification (e.g. dementia, cognitive impairment without dementia, and normal) based on cognitive performance questionnaires has been widely used in population-based studies, providing insight into the population dynamics of dementia. However, researchers have raised concerns about the accuracy of cognitive assessments. MRI and CSF biomarkers may provide improved classification, but the potential improvement in classification in population-based studies is relatively unknown.

### Methods

Data come from the Alzheimer's Disease Neuroimaging Initiative (ADNI). We examined whether the addition of MRI and CSF biomarkers improved cognitive status classification based on cognitive status questionnaires (MMSE). We estimated several multinomial logistic regression models with different combinations of MMSE and CSF/MRI biomarkers. Based on these models, we also predicted prevalence of each cognitive status category using a model with MMSE only and a model with MMSE + MRI + CSF measures and compared them to diagnosed prevalence.

### Results

Our analysis showed a slight improvement in variance explained (pseudo-$R^2$) between the model with MMSE only and the model including MMSE and MRI/CSF biomarkers; the pseudo-$R^2$ increased from .401 to .445. Additionally, in evaluating differences in predicted prevalence for each cognitive status, we found a small improvement in the predicted prevalence of cognitively normal individuals between the MMSE only model and the model with MMSE and CSF/MRI biomarkers (3.1% improvement). We found no improvement in the correct prediction of dementia prevalence.

**Data Availability Statement:** Third party data used in this study was obtained from the the Alzheimer's Disease Neuroimaging Initiative (ADNI) database (https://adni.loni.usc.edu/data-samples/access-data/). The data used in this study can be found in

the dataset entitled 'adnimerge.dta' which can be found accessed through the download study tab, followed by tying in ADNImerge into the search bar, and downloading the zip file called 'ADNIMERGE- Key ADNI Tables merged into one table'. The authors confirm that others would be able to access or request these data in the same manner as the authors. The authors also confirm that they did not have any special access or request privileges that others would not have.

**Funding:** This analysis was supported by funds from the National Institutes of Health (grant numbers T32AG000037; P30 AG043073; P30 AG17265; R00 AG058799; K99AG076964). Data collection and sharing for this project was funded by the Alzheimer's Disease Neuroimaging Initiative (ADNI) (National Institutes of Health Grant U01 AG024904) and DOD ADNI (Department of Defense award number W81XWH-12-2-0012). ADNI is funded by the National Institute on Aging, the National Institute of Biomedical Imaging and Bioengineering, and through generous contributions from the following: AbbVie, Alzheimer's Association; Alzheimer's Drug Discovery Foundation; Araclon Biotech; BioClinica, Inc.; Biogen; Bristol-Myers Squibb Company; CereSpir, Inc.; Cogstate; Eisai Inc.; Elan Pharmaceuticals, Inc.; Eli Lilly and Company; EuroImmun; F. Hoffmann-La Roche Ltd and its affiliated company Genentech, Inc.; Fujirebio; GE Healthcare; IXICO Ltd.;Janssen Alzheimer Immunotherapy Research & Development, LLC.; Johnson & Johnson Pharmaceutical Research & Development LLC.; Lumosity; Lundbeck; Merck & Co., Inc.;Meso Scale Diagnostics, LLC.; NeuroRx Research; Neurotrack Technologies; Novartis Pharmaceuticals Corporation; Pfizer Inc.; Piramal Imaging; Servier; Takeda Pharmaceutical Company; and Transition Therapeutics. The Canadian Institutes of Health Research is providing funds to support ADNI clinical sites in Canada. Private sector contributions are facilitated by the Foundation for the National Institutes of Health (www.fnih.org). The grantee organization is the Northern California Institute for Research and Education, and the study is coordinated by the Alzheimer's Therapeutic Research Institute at the University of Southern California. ADNI data are disseminated by the Laboratory for Neuro Imaging at the University of Southern California. The funders had no role in study design, data collection and analysis, decision to publish, or preparation of the manuscript.

**Competing interests:** The authors have declared that no competing interests exist.

## Conclusion

MRI and CSF biomarkers, while important for understanding dementia pathology in clinical research, were not found to substantially improve cognitive status classification based on cognitive status performance, which may limit adoption in population-based surveys due to costs, training, and invasiveness associated with their collection.

## Introduction

The global burden of dementia is growing due to increased longevity and population aging. Social scientists have sought to document and understand the social differentials and dynamics of dementia in representative population studies. To accomplish this goal, several national studies, which include the Health and Retirement Study (HRS) and its family of studies that now cover about two-thirds of world's population of older people, have integrated cognitive testing. For classification of dementia, these studies have largely relied on algorithms based on cognitive performance assessments such as the Mini-Mental State Exam (MMSE). While these studies have validated their cognitive classification measures against neuropsychological evaluations for select subsamples, recent studies have raised concerns about measurement error in cognitive performance assessments resulting in misclassification that may lead to incorrect conclusions about trends and differences in cognitive performance [1–3]. These measurement concerns have led researchers to consider incorporating biological information in their dementia status classification algorithms to arrive at more accurate classifications [4–6]. However, while research on the contribution of biomarkers for understanding later life cognitive health continues to grow, the potential benefit for population-based surveys regarding cognitive status classification is still relatively unknown.

Recent advancements in Magnetic Resonance Imaging (MRI) present a potential opportunity to improve cognitive status classification. Over the past few decades, researchers have assessed the impact of brain structure on cognitive functioning [7, 8]. Brain structure has been tied to both cognitive functioning and progression of dementia [9–11]. For example, people with Alzheimer's or related dementias are more likely to have smaller hippocampal volume, larger ventricles, and smaller whole brain volume [12–14]. Structural changes in these areas are thought to be linked to brain pathology, which can then lead to cognitive impairment. As such, MRI measures may be a potentially important viable candidate for including in population-based surveys to better classify cognitive status.

Additionally, recent studies have also found significant potential for the incorporation of Cerebrospinal Fluid (CSF) based biomarkers. In particular, several studies have found that Alzheimer's patients had greater levels of P-tau and lower Aß (two hallmark proteins that are widely researched in ADRD pathology). Aß creates amyloid plaques in the brain, while P-tau creates neurofibrillary tangles [15–17]. Both biological changes lead to cognitive impairment by negatively impacting neuronal and axonal connections and communication. While CSF based biomarkers are unlikely to be included in population studies in the foreseeable future due to the associated costs and invasiveness of the procedure, blood-based assays of similar biomarkers are now possible and are much easier to collect for population-based surveys.

We use data from Alzheimer's Disease and Neuroimaging Initiative (ADNI) to assess whether the addition of MRI volumetric measures in key regions of the brain and CSF-based biomarkers (Aß and P-tau) improve cognitive status classification, that is classification as having dementia, cognitive impairment without dementia or being cognitively normal, relative to

cognitive performance examinations normally included in population surveys. We hypothesize that the incorporation of biological information will improve cognitive status classification, which we evaluate by examining the improvement in prediction of cognitive status classification across nested models. We also evaluate the differences between observed prevalence for each cognitive state with these measures included.

## Data and methods

The data used in this study come from the Alzheimer's Disease Neuroimaging Initiative (ADNI) (http://adni.loni.usc.edu/). Started in 2003, ADNI is an interinstitutional initiative that includes 57 clinical sites in the United States and Canada. It has enrolled more than 1,800 older adults 55–90. Its aim is to understand how neuroimaging, fluid and genetic markers, and cognitive assessments can be combined to understand progression into MCI and early AD. ADNI is comprised of a baseline wave with several follow-ups that vary at length but are generally collected every 3 months until progression to severe AD, death, or dropping out from the study. The ADNI data has several phases ADNI 1, ADNI 2, ADNI-Go and ADNI-3 (the current phase). While participants are followed across all phases, participants are added with the start of each new phase to include respondents with varying cognitive statuses. ADNI data are anonymized before access is granted to researchers. Furthermore, projects using ADNI data undergo internal review before access to data is granted and manuscripts must acquire approval before submission to academic journals. Lastly, ADNI studies obtained informed consent from participants prior to data collection.

Our analysis uses participant information from their baseline wave, which could come from any of the four different phases. We have a total of 832 respondents. We restrict our sample to respondents who were not missing on any covariates of interest. About 61% of the sample was excluded (N = 1,343). Most of the missing information came from P-tau (N = 966) and Aß (N = 1,152). Ancillary analysis was conducted to determine whether the associations found with the MRI markers were impacted by the level of missing information from CSF markers: we found no difference in associations and similar magnitudes between both samples.

### Cognitive status diagnosis

Each respondent received a cognitive status diagnosis at baseline interview, which included cognitive normal (CN), significant memory concern (SMC), early mild cognitive impairment (EMCI), late mild cognitive impairment (LMCI), and mild Alzheimer's Disease (AD). We combine these categories to cognitive normal (CN and SMC), cognitive impairment without dementia (EMCI and LMCI), and dementia (AD) to reflect the classification schemas used in large survey studies. ADNI diagnostic criteria come from several different components of the baseline interview: subjective memory information, cognitive performance tests, the clinical dementia rating (CDR), and the presence of limitations of activities of daily living. Respondents with normal memory function on the Logical Memory II subscale of the Wechsler Memory Scale—Revised, a Mini-Mental State Exam (MMSE) score between 24 and 30, a CDR of 0, and no significant impairments in cognitive functions or activities of daily living were considered cognitively normal (CN). Respondents with a subjective memory complaint, abnormal memory function on the Logical Memory II score, an MMSE score between 24 and 30, a CDR of .5 with the Memory Box being .5, but with enough cognitive and functional performance preserved such that a diagnosis of AD could not be made were classified as having cognitive impairment with dementia (CIND). Lastly, respondents with a subjective memory complaint, abnormal memory function on the Logical Memory 11 subscale, an MMSE score between 20 and 26 with some exceptions for respondents with less than 8 years of education who could

have scored less, a CDR rating between .5 and 1, and meeting the NINCDS/ADRDA criteria for probable dementia were classified as having dementia. Each of these components were collected by various study personnel. The cognitive performance questions and memory information were administered by the project interviewer or psychometrician; the CDR was obtained by a CDR rater; for respondents with some form of cognitive impairment, an on-site physician who evaluated all prior information determined whether not the respondent met criteria to be diagnosed with dementia. Furthermore, all respondents regardless of cognitive classification had to have had at least 4 weeks of stable medication routine and not score above the threshold for depression on the geriatric depression scale to be included in the study.

### Mini-Mental State Examination (MMSE) scores

MMSE scores are a frequently used cognitive assessment examination in clinical and survey research. The MMSE has found to have moderately high reliability in assessment of cognitive status and is internally consistent [18]. It has also been found to be sensitive to the severity of dementia.

MMSE scores were obtained at baseline. Following the routine protocol, interviewers collected information on cognitive performance across various domains using tasks including: orientation, memory, recall, naming objects, attention, following verbal and written commands, writing a sentence, and copying a figure. The scores range from 0–30. A higher score is indicative of better cognitive functioning. However, ADNI does not contain many respondents with a sub-20 score. Due to subject selection established by ADNI, respondents with a score of less than 20 were excluded from data collection, unless an exception was made by the physician for respondents with less than 8 years of education (our data contained 2 respondents who scored 19 that were allowed to remain the in study).

### MRI imaging

MRI imaging was collected at baseline for all respondents. However, different scanners were used. A subset of MRIs for ADNI-1 were obtained using 1.5 T scanners, while the rest of ADNI-1, ADNI-GO, ADNI-2, and ADNI-3 used 3 T scanners (for more information, see www.adni.loni.usc.edu). Data from the MRI scans were automatically processed using the FreeSurfer software package (http://sufer.nmr.mgh.harvard.edu) at University of California-San Francisco through the Schuff and Tosun laboratory. For the purposes of this study, we used information on whole brain volume, ventricles and hippocampal volume. These volumetric measures are available within the general ADNI dataset. For ease of interpretability (and after assessing that the variables were normally distributed), we standardized each to have a mean of 0 and a standard deviation of 1.

### CSF markers

CSF markers were collected for all study participants at baseline. Procedures for collecting CSF markers have been previously reported [19]. Aß and p-tau were measured using the multiplex xMAP Luminex platform (Luminex Corporation, Austin, TX) with INNO-BIA AlzBio3 (Innogenetics, Ghent, Beligum) immunoassay kit-based reagents. Information was provided on the number of picograms per milliliter. We standardized these measurements to have a mean of 0 and a standard deviation of 1.

## Other covariates

Information was also collected on gender (male, female), age (54–91), years of education, and APOE gene (0, 1, 2 alleles).

## Analytical strategy

We use used series of multinomial regression models predicting diagnosed cognitive status to assess whether additional covariates improved the variation explained in the model (as indicated by an increase in pseudo-$R^2$). In Model 1, we included covariates for sex, age, education and APOE4 genes (the control model). In Model 2, we add MMSE scores (MMSE only model). In Model 3, we substitute volumetric MRI markers for MMSE (MRI only model). In Model 4, we substitute CSF markers for MRI markers (CSF only model). In Model 5, we include all covariates (MMSE, MRI, and CSF) in a fully saturated model. These models allow us to examine the associations between each set of covariates and cognitive status. From model information, we are able to ascertain whether or not each set of covariates partly explains cognitive status and the potential contribution. For example, by comparing the pseudo-$R^2$ across models, we are able to determine whether certain sets of covariates explain more of the variation in cognitive status than others, and in the fully saturated model, we are able to evaluate the potential improvement in cognitive status classification when all sources of information are included (in our case, we sought to evaluate the improvement in cognitive status classification when MRI and CSF markers were included with MMSE scores: Model 2 compared to Model 5). In addition to these models, ancillary analysis evaluated each covariate separately (a model for each MRI and CSF marker). We found similar patterns to the models with the clusters for each set of markers; therefore, for parsimony, we do not present these findings but instead focus on how the sets of covariates improve cognitive status classification above and beyond only considering cognitive performance examinations.

All models were estimated using STATA 15.1.

## Results

The sample descriptive information is reported in Table 1. The respondents were 73.1 years of age on average. Men constituted 56% of the sample. On average, respondents had 16 years of education. A slight majority had one or two APOE4 alleles: 39.3% had 1 allele and 12.1% had 2 alleles. The average MMSE score was 27.2. And most of the respondents had some form of

**Table 1. Sample characteristics (ADNI, N = 832).**

|  | Mean | (s.d) | % |
|---|---|---|---|
| Age | 73.1 | (7.21) |  |
| Male |  |  | 56% |
| Education (Years) | 16 | (2.72) |  |
| **APOE4** |  |  |  |
| 0 |  |  | 48.6% |
| 1 |  |  | 39.3% |
| 2 |  |  | 12.1% |
| MMSE | 27.2 | (2.67) |  |
| **Cognitive Status** |  |  |  |
| Cognitive Normal |  |  | 19.0% |
| Cognitive Impairment without Dementia |  |  | 60.6% |
| Dementia |  |  | 20.4% |

**Table 2. Coefficients for multinomial logistic regression model predicting cognitive status from MMSE, MRI, and CSF markers (ADNI, N = 832).**

| | Model 1 | Model 2 | Model 3 | Model 4 | Model 5 |
|---|---|---|---|---|---|
| **Dementia** | | | | | |
| Age | 0.0103 | -0.0533* | -0.105*** | -0.0241 | -0.123*** |
| Male | 0.328 | -0.348 | 0.434 | 0.386 | -0.272 |
| Education (Years) | -0.103* | 0.150* | -0.102* | -0.0653 | 0.140* |
| APOE4 Allelles (Reference: 0 Alleles) | | | | | |
| 1 Allele | 1.623*** | 1.278*** | 1.278*** | 0.684* | 0.824* |
| 2 Alleles | 2.729*** | 1.477* | 1.925*** | 1.017* | 0.576 |
| MMSE Score | | -1.782*** | | | -1.633*** |
| Whole Brain Volume | | | 0.340 | | 0.411 |
| Hippocampal Volume | | | -2.006*** | | -1.172*** |
| Ventricle Volume | | | 0.550*** | | 0.373 |
| Ptau | | | | 1.024*** | 0.693*** |
| aß | | | | -1.123*** | -0.261 |
| Constant | -0.160 | 48.68*** | 8.447*** | 2.057 | 50.17*** |
| **Cognitive Impairment Without Dementia** | | | | | |
| Age | -0.0378** | -0.0636*** | -0.0917*** | 0.0562*** | -0.117*** |
| Male | 0.338 | 0.0724 | 0.180 | 0.372 | -0.104 |
| Education (Years) | -0.0242 | 0.0534 | -0.0323 | -0.00917 | 0.0406 |
| APOE Allelles (Reference: 0 Alleles) | | | | | |
| 1 Allele | 0.967*** | 0.809*** | 0.777*** | 0.552* | 0.462 |
| 2 Alleles | 1.349** | 0.879 | 0.815 | 0.663 | 0.192 |
| MMSE Score | | -0.621*** | | | -0.529*** |
| Whole Brain Volume | | | 0.617*** | | 0.580*** |
| Hippocampal Volume | | | -1.254*** | | -1.014*** |
| Ventricle Volume | | | 0.141 | | 0.150 |
| Ptau | | | | 0.586*** | 0.456** |
| aß | | | | -0.182 | -0.0348 |
| Constant | 3.740** | 22.34*** | 8.337*** | 5.218*** | 24.54*** |
| Observations | 832 | 832 | 832 | 832 | 832 |
| Pseudo R-squared | 0.064 | 0.401 | 0.182 | 0.142 | 0.445 |

* p < .05

** p < .01, **p < .001

Source: ADNI data

cognitive impairment: 60.6% had cognitive impairment without dementia and 20.4% had dementia.

We estimated a several multinomial logistic regression models in Table 2 to evaluate the association between each set of covariates with cognitive status and compare the increase in variance explained (pseudo-$R^2$) across the models. In Model 1, the results for the demographic controls and APOE4 alleles are presented. We found that more years of education are associated with a decreased risk of having dementia, and more APOE4 alleles are associated with increased risk. We did not find statistically significant associations for age or gender for dementia. For CIND, we found a positive association with age and APOE4. We did not find statistically significant associations of gender or education for CIND. Overall, the demographic controls and APOE4 explain about 6.4% of the variance in cognitive status (pseudo-$R^2$ = .064).

Next, we added MMSE scores. Corresponding to expectations, higher MMSE scores are associated with a decreased likelihood of having dementia or CIND. The explanatory power of the model also improved markedly. The pseudo-$R^2$ increased from .064 in Model 1 to .401 in Model 2, a 34.6% increase in the explanation of variance with the addition of MMSE to the demographics and APOE4 only model.

In Model 3, we sought to evaluate the association of MRI volumetric measures with controls for demographics and APOE4 with cognitive status. As expected, we found that hippocampal volume is negatively associated with dementia or CIND status: one standard deviation increase in hippocampal volume is associated with an 87% decrease in the likelihood of having dementia and a 71.5% decrease in the likelihood of having CIND (OR .13 and .28, respectively). For dementia only, we found that ventricle volume is positively associated with dementia status. We, however, did not find a statistically significant association with whole brain volume. In contrast, for CIND, we found a positive association for whole brain volume and no statistically significant relationship with ventricle volume. Overall, these models showed a modest improvement in explanation of variance: the pseudo-$R^2$ improved from .064 in Model 1 (control only) to .182 in Model 3 (MRI + Controls).

To evaluate the associations of CSF markers with cognitive status, we examined a model with CSF markers and controls. These results are presented in Model 4. Overall, we find strong associations between CSF markers and dementia (less so for CIND). P-tau has a positive association with CIND and dementia: an increase in P-tau is associated with increased likelihood of having some form of cognitive impairment. For aß, we only find a negative association with dementia (no association was found for CIND): an increase in aß is associated with a decrease in the likelihood of having dementia. Compared to the model with only controls, the inclusion of CSF markers improved the variance explained (.064 in Model 1 to .142 in Model 4). However, compared to earlier models with MMSE and MRI model, the improvement was the smallest.

To evaluate the potential improvements in prediction when including MRI and CSF markers compared to MMSE alone, we evaluated a model with all covariates. Compared to the model with MMSE only (Model 2), including MRI and CSF markers marginally improved the variance explained by 10%: the pseudo-$R^2$ improved from .401 in Model 2 to .445 in Model 5. This provides evidence of marginal improvements in prediction with inclusion of MRI or CSF biomarkers.

Lastly, in Table 3 we evaluated the differences between diagnosed and predicted prevalence from Model 2 (MMSE only) and Model 5 (MMSE+MRI+CSF). These models performed relatively similarly. Both Models 2 and 5 predicted greater levels of CIND (13% for Model 2 and 9.9% for Model 5) and lower levels of dementia (-2.2% for both models) than were diagnosed. The difference between diagnosed and predicted was greatest with CIND and smallest in dementia for both models. Comparing the models to one another, we find small improvements in predicting the correct prevalence across cognitive statuses; the estimates in the fully saturated model were more closely aligned with the diagnosed prevalence, and were driven by

**Table 3. Differences of observed and predicted prevalence of cognitive status in ADNI study (N = 832).**

| Cognitive Status | Diagnosed | Model 2: MMSE Only | Model 4: MMSE + MRI + CSF | Δ between Observed & Predicted from Model 2 | Δ between Observed & Predicted from Model 4 |
|---|---|---|---|---|---|
| Cognitively Normal | 19.0% | 8.2% | 11.3% | -10.8% | -7.7% |
| Cognitive Impairment without Dementia | 60.6% | 73.6% | 70.4% | 13.0% | 9.9% |
| Dementia | 20.4% | 18.3% | 18.3% | -2.2% | -2.2% |

improvements in cognitively normal and CIND. Lastly, we also evaluated specificity and sensitivity of models for individuals. We found that correctly classified respondents only improved by 2%: Model 2 correctly predicted 71% of individuals and Model 5 predicted 73% of individuals (see S1 Table). This further provides evidence that adding MRI and CSF measures do not greatly improve prevalence estimates for dementia from models that use MMSE alone.

## Discussion

The growing global burden of dementia has led social science researchers to develop measurements to document the changing trends and disparities of cognitive impairment in large survey samples. Researchers have raised concerns about misclassification found in cognitive status algorithms based on cognitive performance assessments, which may have a negative impact on the ability of researchers to accurately evaluate the population dynamics of cognitive impairment [2, 3, 20]. Recent advancements in MRI and other biomarkers have led researchers to question whether biomarker information may improve the algorithms used in cognitive status classification. Our study used a clinically based sample that collected the MMSE, MRI and CSF biomarker information along with doctor diagnosed cognitive status to examine whether additional MRI and CSF marker information improved upon cognitive status classification based on cognitive performance measures.

We found minimal evidence that MRI or CSF biomarkers improved upon cognitive status classification after cognitive performance measures were included. While MRI and CSF biomarkers explained some of the variance in cognitive status (as shown in Models 3 & 4), when included with MMSE (Model 5), the variance explained increased from 40% to 44% (a 10% relative increase). This finding was further illustrated by the small improvements made from predicted prevalence and diagnosed prevalence in Table 3. This suggests that while MRI and CSF measures are important pieces of information used to understand the underlying pathology of ADRD as clinical research has shown [16, 21–24], its benefits for improving classification of cognitive status in large population based surveys may be more limited. Cognitive performance measures are already widely used across a host of national surveys. These cognitive performance examinations can be performed over the phone (such as the TICS score in the HRS), which would not require thousands of participants to come into a clinic as would be required with MRI and CSF collection. MRI may hinder participation because of small, enclosed space that may make individuals uncomfortable. CSF collection is invasive in that it requires a spinal tap. Both procedures require proper medical equipment and trained personnel to collect and process biological information, resulting in significantly greater costs than is required for the administration of cognitive performance examinations. Therefore, given the lower cost and large explanatory power of cognitive performance examinations, coupled with minimal improvements with MRI and CSF biomarker information, cognitive performance examinations may continue to be the sole component of cognitive status evaluation in large scale studies that are seeking documentation of trends and inequalities in cognitive impairment status.

While the results of the study do not provide strong evidence that MRI or CSF biomarkers should be incorporated into large scale surveys to improve cognitive status classification, the search for other biomarkers may be more fruitful (especially blood-based markers that are less costly and easier to collect and process). In recent years, scientific advancements have been made to better understand the relationship between blood-based biomarkers and dementia [25–27]. Blood-based biomarkers are promising in that they are easier and less costly to collect. While the ADNI study does collect some information on blood-based biomarkers, the collection has been limited in both sample and scope. In our study, we performed ancillary analysis that evaluated the plasma levels of aß, P-tau, and neurofilament light (see S2 Table). We found

no evidence that aß or P-tau plasma levels were associated with cognitive status. We also found correlations below .1 between the plasma and CSF levels of aß and P-tau, which is similar to findings in other studies using aß40 [28, 29]. However, recent advancements in blood-based biomarker assays may provide a potential avenue in which to improve cognitive status classification in large scale surveys can be improved; this should be further investigated as research advances in this area.

Additionally, it is also important to note that while we find minimal improvements in cognitive status classification with the inclusion of MRI and CSF biomarkers, these markers may be more important for other types of scientific questions that will push cognitive health research forward. For example, understanding the underlying pathology for cognitive impairment in the population may provide greater insight into potential treatments that will improve the health and wellbeing of individuals and can have implications for understanding health inequities. Contributions from these types of studies cannot be understated. Therefore, in considering whether to add these MRI and CSF markers to population-based surveys that often collect information on several thousand individuals, researchers will have to determine the purpose and use of the markers beyond cognitive status classification, whether population-based samples can add to scientific insight beyond existent or emerging community or clinical-based studies, and the extent to which their studies will participate (such as a targeted group or randomized subsample).

Our study includes several limitations. First, the sample was composed of mostly well-educated, white older adults. This limitation impacts the ability to generalize to the larger US population (and to other countries). For example, studies have found that people with greater levels of education have higher cognitive functioning, regardless of brain pathology [30, 31]. Therefore, well-educated adults may be less sensitive to brain pathology than those with fewer years of education. As a result, MRI and CSF biomarkers may have better explanatory power in adults with fewer years of education. But the extent to which they would improve upon cognitive status classification from cognitive performance questionnaires is unknown. Future studies should assess these potential contributions for adults with fewer years of education. Second, due to selection criteria (an MMSE of over 19), ADNI only includes older adults with higher levels of cognitive functioning. Therefore, we cannot evaluate how MRI and CSF biomarkers may improve upon cognitive performance questionnaires to classify cognitive status for people with low levels of cognitive performance on the exams. It may be that cognitive performance questionnaires are an even stronger predictor of cognitive status (ADRD in this case) among those with lower cognitive functioning because these cases are more clear-cut in terms of classification, leading to an even smaller contribution from MRI and CSF biomarkers above and beyond cognitive performance tests.

## Conclusion

Accurate cognitive status classification in large-scale surveys across the world is fundamental to understanding the population health dynamics of dementia, whose findings will have broad implications for government policy. Recent debates among scientists have emerged that seek to improve the precision and sensitivity of the algorithms used to evaluate the burden of cognitive impairment in the population. Some have proposed to incorporate more direct measurements of brain pathology. Our study shows that measurements of MRI and CSF biomarkers do little to improve upon the explanatory power of cognitive performance tests, perhaps indicating that large national surveys should continue to incorporate cognitive performance as their primary means of assessing cognitive status and seek to improve upon existing algorithms as these cognitive functioning measurements also adapt and change to the various outcomes and populations on interest.

## Supporting information

**S1 Table. Specificity and sensitivity of cognitive status classification model with MMSE only and MMSE + MRI + CSF predicting diagnosed cognitive status.**
(DOCX)

**S2 Table. Multinomial regression model predicting cognitive status at baseline with plasma marks of P-Tau and Aß.**
(DOCX)

## Author Contributions

**Conceptualization:** Mateo P. Farina, Joseph Saenz, Eileen M. Crimmins.

**Data curation:** Mateo P. Farina.

**Formal analysis:** Mateo P. Farina.

**Methodology:** Eileen M. Crimmins.

**Writing – original draft:** Mateo P. Farina.

**Writing – review & editing:** Mateo P. Farina, Joseph Saenz, Eileen M. Crimmins.

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
