## [Decision Letter · Decision Letter 0]

3 Mar 2023

PONE-D-23-00710Does Adding MRI and CSF-Based Biomarkers Improve Cognitive Status Classification based on Cognitive Performance Questionnaires?PLOS ONE

Dear Dr. Farina,

Thank you for submitting your manuscript to PLOS ONE. After careful consideration, we feel that it has merit but does not fully meet PLOS ONE’s publication criteria as it currently stands. Therefore, we invite you to submit a revised version of the manuscript that addresses the points raised during the review process.

We look forward to receiving your revised manuscript.

Kind regards,

Kensaku Kasuga

Academic Editor

PLOS ONE

Journal Requirements:

4. Please ensure that you include a title page within your main document. You should list all authors and all affiliations as per our author instructions and clearly indicate the corresponding author.

Additional Editor Comments:

Please revise your manuscript according to the reviewers' suggestions and provide a point-by-point response to the reviews.

Reviewers' comments:

Reviewer's Responses to Questions

**Comments to the Author**

1. Is the manuscript technically sound, and do the data support the conclusions?

Reviewer #1: Yes

Reviewer #2: No

Reviewer #3: Yes

2. Has the statistical analysis been performed appropriately and rigorously? 

Reviewer #1: Yes

Reviewer #2: Yes

Reviewer #3: Yes

3. Have the authors made all data underlying the findings in their manuscript fully available?

Reviewer #1: Yes

Reviewer #2: Yes

Reviewer #3: Yes

4. Is the manuscript presented in an intelligible fashion and written in standard English?

Reviewer #1: Yes

Reviewer #2: Yes

Reviewer #3: Yes

5. Review Comments to the Author

Reviewer #1: Authors reported, MRI and CSF biomarkers, while important for understanding dementia pathology in clinical research, were not found to substantially improve cognitive status classification based on cognitive status performance, which may limit adoption in population-based surveys due to costs, training, and invasiveness associated with their collection.

This is well conducted study.

Reviewer #2: As imaging, cerebrospinal fluid, and blood biomarkers are achieving more attention in the diagnosis of dementia, this is an important study that reaffirms the importance of assessing cognitive performance in the diagnosis of dementia and cognitive impairment. However, I have a major concern.

Authors mention that MRI and CSF biomarkers, while important for understanding dementia pathology in clinical research, were not found to substantially improve cognitive status classification based on cognitive status performance. First of all, isn't that a given? Unless, overuse of biomarkers would interfere with classification of cognitive status, imaging and CSF biomarkers are necessary to investigate the "causes" of cognitive impairment, not to detect the "presence" of cognitive impairment. For example, the presence of Alzheimer's pathology in the brain does not necessarily result in Alzheimer's dementia. Without cognitive performance tests and assessment of ADLs, it is impossible to determine if dementia is present in Alzheimer's disease. Even in the diagnostic criteria for Alzheimer's disease dementia, no imaging or CSF biomarkers are needed to diagnose “probable AD dementia”. On the other hand, once the cognitive impairment is observed, imaging, CSF, and blood biomarkers are useful to determine whether it is due to the accumulation of specific proteins, or associated with cerebrovascular disease, or whether it is a treatable dementia such as iNPH and epilepsy.

Secondly, authors conclude that MRI and CSF biomarkers, which may limit adoption in population-based surveys due to the cost, training, and invasiveness associated with their collection. However, for example, this study did not examine the cost-effectiveness of using imaging and CSF biomarkers in population-based surveys. Understanding the underlying pathology of dementia for the appropriate diagnosis and medication, such as use of disease-modifying drugs, may allow patients to stay in their homes longer and reduce total health care costs. Furthermore, reviewer disagree with the indiscriminate implementation of invasive testing. However, future search for non-invasive and inexpensive biomarkers will have greater benefits, but for development, comparisons with currently established invasive biomarkers are needed.

Reviewer #3: This paper systematically investigated the classification of Cognitive status (Dementia, cognitive impairment without dementia and normal) using multinomial logistic regression. The regression is performed with different models on the basis of combination of covariates. The study is primarily focused on the contribution of MRI and CSF biomarkers to classification of cognitive status.

I have some concerns on this paper.

1) The paper is well written and the finding of the paper is clearly described.

2) The classification is done on the basis of odd ratio using multinomial logistic regression. Results could be presented graphically so that the patterns, trends and relationships between different variables could be easily understood.

3) Numerous papers are published on Cognitive status classification using the biomarkers (MRI, CSF) used in this paper. Only the difference is, majority of works classify on the basis of probability and the proposed method classified based on odd ratio. What is the major contribution in this paper besides expressing the classification in terms of odds ratio?

6. PLOS authors have the option to publish the peer review history of their article (what does this mean?). If published, this will include your full peer review and any attached files.

Reviewer #1: No

Reviewer #2: No

Reviewer #3: No

---

## [Author Response · Author response to Decision Letter 0]

24 Mar 2023

We thank the editor and reviewers for the opportunity to improve the manuscript. We were pleased with the overall positive assessment of the manuscript, and in response to the reviewers’ comments, we have amended manuscript to further clarify our contribution and have highlighted the importance of MRI and CSF biomarkers for research on brain pathology. We believe that the reviewers have helped us improve our contribution. Our response to each point and manuscript edits are detailed below: 

Reviewer #1: Authors reported, MRI and CSF biomarkers, while important for understanding dementia pathology in clinical research, were not found to substantially improve cognitive status classification based on cognitive status performance, which may limit adoption in population-based surveys due to costs, training, and invasiveness associated with their collection.

This is well conducted study.

We thank the reviewer for their consideration of the manuscript and appreciate their positive remarks. 

Reviewer #2: As imaging, cerebrospinal fluid, and blood biomarkers are achieving more attention in the diagnosis of dementia, this is an important study that reaffirms the importance of assessing cognitive performance in the diagnosis of dementia and cognitive impairment. However, I have a major concern.

Authors mention that MRI and CSF biomarkers, while important for understanding dementia pathology in clinical research, were not found to substantially improve cognitive status classification based on cognitive status performance. First of all, isn't that a given? Unless, overuse of biomarkers would interfere with classification of cognitive status, imaging and CSF biomarkers are necessary to investigate the "causes" of cognitive impairment, not to detect the "presence" of cognitive impairment. For example, the presence of Alzheimer's pathology in the brain does not necessarily result in Alzheimer's dementia. Without cognitive performance tests and assessment of ADLs, it is impossible to determine if dementia is present in Alzheimer's disease. Even in the diagnostic criteria for Alzheimer's disease dementia, no imaging or CSF biomarkers are needed to diagnose “probable AD dementia”. On the other hand, once the cognitive impairment is observed, imaging, CSF, and blood biomarkers are useful to determine whether it is due to the accumulation of specific proteins, or associated with cerebrovascular disease, or whether it is a treatable dementia such as iNPH and epilepsy.

Secondly, authors conclude that MRI and CSF biomarkers, which may limit adoption in population-based surveys due to the cost, training, and invasiveness associated with their collection. However, for example, this study did not examine the cost-effectiveness of using imaging and CSF biomarkers in population-based surveys. Understanding the underlying pathology of dementia for the appropriate diagnosis and medication, such as use of disease-modifying drugs, may allow patients to stay in their homes longer and reduce total health care costs. Furthermore, reviewer disagree with the indiscriminate implementation of invasive testing. However, future search for non-invasive and inexpensive biomarkers will have greater benefits, but for development, comparisons with currently established invasive biomarkers are needed.

We thank the reviewer for the comments that highlight the importance of biomarkers for research, especially as it pertains to pathology. To address the first concern the reviewer brings up about whether it is a given that dementia pathology markers do not improve cognitive status classification based on cognitive performance, we believe that this area of debate remains open. Scientific surveys, especially large nationally representative studies, are trying to determine how best to improve cognitive status classification, especially as it becomes an increasingly important area of research. Ongoing debates are being held about the influence of practice effects and cultural differences that impact exam results using current practices in surveys (Feeney et al., 2016; Gianattasio et al., 2019; Hale et al., 2020). Also, there is room for cognitive status classification to improve. Some of the best performing measures correctly classify respondents have a sensitivity that ranged from 18% to 62%, indicating there is remove from improvement in correctly classifying people with cognitive impairment (Gianattasio et al., 2019) . Therefore, we believe that these questions are not settled, and that this manuscript contributes to this conversation.

One area of significant debate is whether and how to incorporate brain pathology markers to improve cognitive status classification in addition to cognitive performance questionnaires. We agree with the reviewer that dementia classification necessitates the inclusion of poor cognitive functioning—the opposite has not been argued in the manuscript. In fact, our introduction and analytical approach centers on the importance of cognitive functioning markers by focusing on the use of cognitive functioning tests for classification of cognitive status in studies. We then add brain pathology biomarkers to observe whether the classification based on cognitive functioning is improved. We recognize that imaging, CSF, and blood-based biomarkers are important for understanding pathology. Our conclusion is not to discourage their collection but rather to clarify how best to use them and when to incorporate them in population-based studies and that if they are used for improvements of cognitive status classification then their application may be more limited. We do not believe that the positions are in opposition. To address the reviewer’s concerns, we have amended the manuscript to recognize the importance that research to improve understanding of the underlying pathological developments which put people at risk for dementia will have large and important for individuals, communities, and populations. We have added text to the manuscript as follows: 

“Additionally, it is also important to note that while we find minimal improvements in cognitive status classification with the inclusion of MRI and CSF biomarkers, these markers may be more important for other types of scientific questions that will push cognitive health research forward. For example, understanding the underlying pathology for cognitive impairment in the population may provide greater insight into potential treatments that will improve the health and wellbeing of individuals and can have implications for health inequities. Contributions from these types of studies cannot be understated. Therefore, in considering whether to add these MRI and CSF markers to population-based surveys that often collect information on several thousand individuals, researchers will have to determine the purpose and use of the markers beyond cognitive status classification, whether population-based samples can add to scientific insight beyond existent or emerging community or clinical-based studies, and the extent to which their studies will participate (such as a targeted group or randomized subsample).” (pg.8)

Reviewer #3: This paper systematically investigated the classification of Cognitive status (Dementia, cognitive impairment without dementia and normal) using multinomial logistic regression. The regression is performed with different models on the basis of combination of covariates. The study is primarily focused on the contribution of MRI and CSF biomarkers to classification of cognitive status.

I have some concerns on this paper.

1) The paper is well written and the finding of the paper is clearly described.

2) The classification is done on the basis of odd ratio using multinomial logistic regression. Results could be presented graphically so that the patterns, trends and relationships between different variables could be easily understood.

We thank the reviewer for this suggestion. We have debated and considered how best to incorporate reviewer feedback as it pertains to Table 2. However, we have not found a simple solution that would not lead to potential confusion. Usually, graphical representations of coefficient changes are used in model series that add additional covariates in a subsequent manner to determine how specific coefficients change based on the inclusion. In this manuscript, the emphasis has been placed more or lesson on the changes in explanatory power of each model as characterized by the R2 and covariates are not added subsequentially. Therefore, we believe that a graphical representation may obfuscate the main purpose of the models. 

3) Numerous papers are published on Cognitive status classification using the biomarkers (MRI, CSF) used in this paper. Only the difference is, majority of works classify on the basis of probability and the proposed method classified based on odd ratio. What is the major contribution in this paper besides expressing the classification in terms of odds ratio?

We thank the reviewer for the question about the contribution. We have amended the manuscript to highlight our contribution so that it will remain clear for future readers. Briefly, our contribution is not the use of odds ratios to predict cognitive status classification. Rather the contribution is considering whether cognitive status classification based on cognitive functioning tests improves with additional MRI and CSF biomarkers. To the best of knowledge, this question has not been addressed before. Instead, research has largely observed them separately, considering their individual associations with cognitive impairment. 

Feeney, J., Savva, G. M., O’Regan, C., King-Kallimanis, B., Cronin, H., & Kenny, R. A. (2016). Measurement Error, Reliability, and Minimum Detectable Change in the Mini-Mental State Examination, Montreal Cognitive Assessment, and Color Trails Test among Community Living Middle-Aged and Older Adults. Journal of Alzheimer’s Disease: JAD, 53(3), 1107–1114. https://doi.org/10.3233/JAD-160248

Gianattasio, K. Z., Wu, Q., Glymour, M. M., & Power, M. C. (2019). Comparison of Methods for Algorithmic Classification of Dementia Status in the Health and Retirement Study. Epidemiology (Cambridge, Mass.), 30(2), 291–302. https://doi.org/10.1097/EDE.0000000000000945

Hale, J. M., Schneider, D. C., Gampe, J., Mehta, N. K., & Myrskylä, M. (2020). Trends in the Risk of Cognitive Impairment in the United States, 1996–2014. Epidemiology (Cambridge, Mass.), 31(5), 745–754. https://doi.org/10.1097/EDE.0000000000001219

---

## [Decision Letter · Decision Letter 1]

18 Apr 2023

Does adding MRI and CSF-based biomarkers improve cognitive status classification based on cognitive performance questionnaires?

PONE-D-23-00710R1

Dear Dr. Farina,

We’re pleased to inform you that your manuscript has been judged scientifically suitable for publication and will be formally accepted for publication once it meets all outstanding technical requirements.

Kind regards,

Kensaku Kasuga

Academic Editor

PLOS ONE

Additional Editor Comments (optional):

All issues I proposed have been addressed. The manuscript deserves to be published.

Reviewers' comments:

Reviewer's Responses to Questions

**Comments to the Author**

1. If the authors have adequately addressed your comments raised in a previous round of review and you feel that this manuscript is now acceptable for publication, you may indicate that here to bypass the “Comments to the Author” section, enter your conflict of interest statement in the “Confidential to Editor” section, and submit your "Accept" recommendation.

Reviewer #1: All comments have been addressed

Reviewer #2: All comments have been addressed

Reviewer #3: All comments have been addressed

2. Is the manuscript technically sound, and do the data support the conclusions?

Reviewer #1: Yes

Reviewer #2: Yes

Reviewer #3: Yes

3. Has the statistical analysis been performed appropriately and rigorously? 

Reviewer #1: Yes

Reviewer #2: Yes

Reviewer #3: Yes

4. Have the authors made all data underlying the findings in their manuscript fully available?

Reviewer #1: Yes

Reviewer #2: Yes

Reviewer #3: Yes

5. Is the manuscript presented in an intelligible fashion and written in standard English?

Reviewer #1: Yes

Reviewer #2: Yes

Reviewer #3: Yes

6. Review Comments to the Author

Reviewer #1: authors revised manuscript well.

this manuscript described about MRI and CSF-based biomarkers improve cognitive status classification.

this is well conducted study.

Reviewer #2: (No Response)

Reviewer #3: The author has successfully addressed all of the previously identified issues, resulting in a noticeable improvement in the overall quality of the paper.

7. PLOS authors have the option to publish the peer review history of their article (what does this mean?). If published, this will include your full peer review and any attached files.

Reviewer #1: No

Reviewer #2: No

Reviewer #3: No

---

## [Editor Report · Acceptance letter]

28 Apr 2023

PONE-D-23-00710R1 

Does adding MRI and CSF-based biomarkers improve cognitive status classification based on cognitive performance questionnaires? 

Dear Dr. Farina:

I'm pleased to inform you that your manuscript has been deemed suitable for publication in PLOS ONE. Congratulations! Your manuscript is now with our production department. 

Kind regards, 

on behalf of

Dr. Kensaku Kasuga 

Academic Editor

PLOS ONE